# Assessment of Knowledge and Attitude of Dental Students towards HIV and Its Oral Manifestations in Saudi Arabia—A Cross-Sectional Study

**DOI:** 10.3390/healthcare10081379

**Published:** 2022-07-25

**Authors:** Faisal Mehsen Alali, Bassel Tarakji, Abdullah Saad Alqahtani, Nasser Raqe Alqhtani, Abdullah Bin Nabhan, Adel Alenzi, Ali Alrafedah, Ali Robaian, Mohammed Noushad, Omar Kujan, Abdullah Alshehri, Ibrahim Saleh Aljulayfi, Mohammad Zakaria Nassani

**Affiliations:** 1Department of Oral and Maxillofacial Surgery and Diagnostic Sciences, College of Dentistry, Prince Sattam Bin Abdulaziz University, Al Kharj 11942, Saudi Arabia; f.alali@psau.edu.sa (F.M.A.); n.alqhtani@psau.edu.sa (N.R.A.); a.binnabhan@psau.edu.sa (A.B.N.); a.alenazi@psau.edu.sa (A.A.); a.alrafedah@psau.edu.sa (A.A.); 2Department of Preventive Dental Sciences, College of Dentistry, Prince Sattam Bin Abdulaziz University, Al Kharj 11942, Saudi Arabia; ab.alkahtani@psau.edu.sa; 3Department of Conservative Dental Sciences, College of Dentistry, Prince Sattam Bin Abdulaziz University, Al Kharj 11942, Saudi Arabia; ali.alqahtani@psau.edu.sa (A.R.); am.alshehri@psau.edu.sa (A.A.); 4Department of Restorative and Prosthetic Dental Sciences, College of Dentistry, Dar Al Uloom University, Riyadh 13314, Saudi Arabia; inya113@yahoo.com (M.N.); mznassani@dau.edu.sa (M.Z.N.); 5UWA Dental School, The University of Western Australia, Nedlands, WA 6009, Australia; omar.kujan@uwa.edu.au; 6Department of Prosthetic Dental Sciences, College of Dentistry, Prince Sattam Bin Abdulaziz University, Al Kharj 11942, Saudi Arabia; i.aljulifi@psau.edu.sa

**Keywords:** knowledge, attitude, oral manifestation, HIV

## Abstract

Background: It is essential for practicing dentists to have adequate knowledge of HIV/AIDS and its implications in the dental practice. Their attitudes should also be up to the professional expectations. This study aimed to assess knowledge of HIV/AIDS infection and attitudes towards its patients among dental students in Saudi Arabia. Methods: A questionnaire was prepared and distributed among a sample of dental students in Saudi Arabia. Twenty questions related to knowledge, attitudes, and oral manifestation of HIV were presented. Results: A total of 405 questionnaires were completed (67% response rate). Participants showed low knowledge of safety regarding HIV (39.5%) and 44.2% indicated that they would take an HIV test after needle stick injury. The proportion of correct answers regarding transmission of HIV through saliva, cardiopulmonary resuscitation, and aerosols by hand pieces was 41%, 37.5%, and 26.4%, respectively. Almost 50% of the surveyed dental students expressed unwillingness to treat HIV-positive patients. Poor knowledge was noted regarding oral manifestations of HIV (32.7%). Of the participants, 57.8% indicated that infection control procedures are very important for the treatment of HIV patients and 50.6% believed that a dentist can make a decision to reject the treatment of HIV patients. According to the participating dental students, the three major oral manifestations in HIV patients are hairy leukoplakia (47.4%), oral candidiasis (44.7%), and Kaposi’s sarcoma (43.5%). Conclusion: Inadequate knowledge and unprofessional attitude towards HIV/AIDS patients were identified among this group of dental students in Saudi Arabia. Dental educators and health care planners in Saudi Arabia should plan to promote the knowledge and attitudes of dental students in Saudi Arabia towards the treatment of HIV patients.

## 1. Introduction

AIDS is a serious systemic disease, and it is no longer a rare illness [1,2]. Commonly, the mouth is one of the main sites where the signs of AIDS appear, so the dentist may be the first to discover, diagnose, and deal with this ailment. Management and treatment of the AIDS patient require a careful and proper approach for maintaining the health of the doctor and the auxiliary team, and prevention of disease transmission [3,4]. Therefore, it is necessary to build a diagnostic mentality among dental students, and provide them with sufficient skills and knowledge to recognize the oral manifestations of AIDS and deliver the appropriate management for this type of patient [5].

We have to admit that the dental treatment of AIDS patients can be problematic and the attitude of the dentist or dental student may play a significant role in this respect. Additionally, the dentist’s fear of treating AIDS patients, and/or lack of/insufficient knowledge on the management of such patients may affect the outcome of dental treatment [6].

Research findings show that dental management of AIDS patients is a controversial issue, and that opinions and attitudes about the dental treatment of AIDS patients vary among dentists across different countries [7,8,9,10]. It was found in a Malaysian study that most of the dental students had a good level of knowledge of AIDS and its signs, but their attitude towards its treatment was not associated with that knowledge [11]. Similarly, in a Chinese study, dental students had a good knowledge of AIDS, but showed unprofessional attitude toward its treatment [12]. In addition, in the Kingdom of Saudi Arabia, in Jazan, the attitude towards treating AIDS patients among dental students was not up to the professional standard and their related knowledge was low [13]. However, in another Saudi study, in Riyadh, the dental students showed good knowledge and attitude towards HIV patients and its treatment [14]. The prevalence of HIV in Saudi Arabia and in the Middle East is considered to be among the lowest rates globally (0.02%) [15]. However, dental students should be well-prepared and educated for the management of AIDS patients. Usually, AIDS patients seek dental treatment due to the fact that oral manifestations of HIV infection are very common and this is known to be a strong indication of disease progression [16]. Oral lesions can be encountered in 30–80% of AIDS/HIV patients and this is still a major health problem worldwide [16]. With the increasing numbers of people with HIV/AIDS receiving oral and dental care, dental students and dentists should have sufficient knowledge of the disease, and their attitude should meet the professional expectations [17]. HIV- and AIDS-related knowledge among dental students provides a crucial foundation for the efforts aimed at developing appropriate knowledge and attitude among dental students towards disease in the dental clinic [18]. Standard dental/oral care requires specific strategies to prevent disease transmission among oral health care workers, and from patient to patient [19]. An essential step in building such strategies is to identify the current level of knowledge, attitude, and perception of the oral health care providers who hold the main responsibility for the provision of quality oral care for all patients including the HIV patients. Unfortunately, there are few studies available evaluating the knowledge and attitudes of dental students about HIV/AIDS in Saudi Arabia. The aim of this study was to determine dental students’ responses to safety-related questions about HIV, attitude towards treatment of HIV-positive patients, and knowledge of oral manifestation of HIV patients, and to compare the findings on the basis of gender and study level in dentistry.

## 2. Materials and Methods

### 2.1. Study Design

This was a cross-sectional questionnaire-based study.

### 2.2. Setting and Sample

The targeted sample of this study was undergraduate dental students at government and private universities in Saudi Arabia including first, second, third, fourth, fifth, and sixth years in addition to internship dental students in the 2020–2021 academic year. Sample size was calculated by using the Open-Source Epidemiologic Statistics for Public Health software—OpenEpi. We used 50% as the hypothesized % frequency of outcome factor in the population which is recommended for unknown frequency and 5% as absolute precision. The results were 384 for a 95% confidence interval. We added 5% to overcome the possibilities of missing data, so the required minimum sample size was 403 to achieve 95% power.

The questionnaire was developed based on previous similar studies [7,10,11,12,13,20,21]. At the planning stage, the survey was administered to three experts in this field who had a strong background in HIV. Based on their feedback, the questionnaire was revised accordingly. The validation of the questionnaire was further tested by a pilot study among a group of dental students (*n* = 30), then modified. The questionnaire consisted of four sections: the first section sought information regarding demographics of the participants including gender, age, and academic study year. The second section comprised 5 items to evaluate students’ knowledge about the safety precautions in dealing with HIV dental patients. The third section included 6 statements to assess attitudes of the dental students toward the treatment of HIV patients. The fourth section was designed to determine knowledge of dental students regarding oral manifestation of HIV.

Afterwards, a Google Form of the survey was developed and distributed among the targeted random sample of dental students by social media and WhatsApp groups over 6 months (September 2020 to March 2021). The authors followed a convenience random sampling protocol in recruiting participants. Due to COVID-19 restrictions, online administration of the survey was the best possible option.

The introduction of the questionnaire presented an explanation on the purpose of study and the participants were assured that all the collected information would be treated confidentially and anonymously. Furthermore, six reminders were sent through WhatsApp groups and other social media platforms aiming to increase the response rate. Responses to the questionnaire were sensitive to the internet protocol (IP) address, ensuring no duplicated responses. Informed consent was taken from all participants before the completion of the survey. The ethical approval was obtained from the Ethical Review Board (REC-HSD-88-2021), Prince Sattam Bin Abdulaziz University, Al Kharj, Saudi Arabia.

### 2.3. Data Analysis

Data analysis was carried out with the SPSS statistical package (IBM SPSS Statistics for Windows, Version 20.0, Released 2011, IBM Corp., Armonk, NY, USA). Characteristics of participating dental students were presented by descriptive statistics and their responses were illustrated by frequency tables. The chi-square statistic was used to assess any possible association between questionnaire items and gender of the students and study level. A *p*-value < 0.05 was considered significant.

## 3. Results

### 3.1. Characteristics of Participants

In order to achieve the target sample size and taking into account the expected high number of non-respondents, 600 dental students studying in Saudi Arabia were contacted. Among these, 405 completed the questionnaire (67% response rate), which included 269 male (66.4%) and 136 (33.6%) female participants as shown in Table 1. The study sample comprised all academic years for a random sample of dental students in the Kingdom of Saudi Arabia, starting from the first to the sixth year, in addition to internship students. Participants were divided into three study levels: the first is preclinical, 220 (54.3%), including first, second, and third year, the second is clinical, 130 (32.1%), including fourth, fifth, and sixth year, whereas the third is internship students, 55 (13.6%). The study population involved Saudi dental undergraduate students of both genders and different ages. The age of 71% of the respondents was between 18 and 23 years, and 23–30 years for 29% of the participants. The proportion of male students exceeded the females (66.4% compared to 33.6%). Table 1 shows characteristics of participants.

### 3.2. Safety-Related Questions about HIV by Gender and Study Level

Our results showed that the overall knowledge of the dental students regarding safety related to HIV was 39.5% (Table 2). Regarding knowledge of HIV tests after needle stick injury, no statistical differences were identified among the dental students from different study levels nor between male and female students (*p* > 0.05) (Table 2). Similarly, there were no significant statistical differences between the three study levels regarding transmission of the virus by saliva, cardiopulmonary resuscitation, or aerosols of hand pieces (*p* > 0.05). Notably, the highest percentage of correct answers for the transmission of HIV through saliva and cardiopulmonary resuscitation was identified among students in the clinical level (43.1%, 42.3%, respectively), whereas the highest percentage of correct answers for the potential transmission of HIV through aerosols of the hand pieces was among dental intern students (32.7%). When asked about the validity of ELISA tests in the detection HIV infection, the intern students significantly showed the best level of knowledge compared to clinical and preclinical students (74.5% compared to 66.2% and 31.4%). Likewise, female dental students showed a higher correct response level (59.6%) compared to male students (42.8%) regarding ELISA as a trusted test for HIV infection (*p* < 0.05). Additionally, female students exhibited better knowledge (44.9%) compared to male students (33.8%) regarding transmission of the HIV through cardiopulmonary resuscitation (*p* < 0.05) (Table 2).

### 3.3. Attitude-Related Questions Regarding Treatment of HIV Patients by Gender and Study Level

The results revealed clear association between study level and attitude-related questions with the preclinical dental students showing a less positive attitude towards treatment of HIV patients compared to clinical and internship students (*p* < 0.05) (Table 3). Moreover, attitudes of female students were more positive regarding the following statements: it is an ethical issue for an HIV-positive patient to be treated by a dentist, infection control procedures are very important for the treatment of HIV patients and take a long time and may affect the work quality of the dentist, routine dental care should be a part of the treatment of patients with HIV and the dentist can make a decision to refuse to treat an HIV-positive patient (*p* < 0.05). However, no significant difference was identified between male and female students regarding the acceptance to treat HIV-positive patients, and all dental patients must be treated even if they have HIV/AIDS (*p* > 0.05) (Table 3).

### 3.4. Knowledge-Related Questions Regarding Oral Manifestation of HIV by Gender and Study Level

On the level of study population, a less than adequate level of knowledge regarding oral manifestation of HIV can be recognized (32.7%, overall percentage of correct answers). However, the internship dental students showed the best level of knowledge compared to clinical and preclinical students (49.7% compared to 43.5% and 22.1%). Similarly, no association was identified on the level of study sample between gender and participants’ knowledge regarding oral manifestation of HIV (*p* > 0.05) (Table 4).

## 4. Discussion

Most of the published papers in Saudi Arabia regarding evaluation of knowledge and attitude of dental students towards HIV patients involved only students at the clinical study level, while preclinical and intern students were overlooked. To the authors’ knowledge, this is the first study in Saudi Arabia that involved dental students in all study years starting from the preclinical level and including clinical study level and intern students. Dental students at Saudi dental schools might have comprehensive knowledge of HIV and its oral manifestation due to studying oral pathology at the preclinical study level and oral medicine at the clinical study level.

Our results showed that overall knowledge of dental students regarding safety-related questions about HIV is poor (39.5%). Grover et al. [22], Yildirim et al. [23], and Singh et al. [11] indicated that knowledge of dental students about HIV patients was less than adequate (54%, 56.26%, and 72.7%, respectively). Although our study showed a lower level of knowledge in comparison with the abovementioned studies, this study included all study levels of dental students in preclinical, clinical, and intern levels, whereas other studies included clinical study level or clinical and preclinical only [11,22,23]. Our study showed that knowledge of clinical and intern students was higher than the preclinical students, indicating an increase in knowledge with advancing level.

Although studies have shown a very rare association between needle stick injury or cut exposures to HIV-contaminated blood and infection with the virus (0.3%) [13], such wounds may be perceived as a high-risk factor for HIV infection among dentists [14,22]. Our results showed that 48.5% of students at the clinical study level would test for HIV in the event of needle stick injury, which is similar to the results (49%) of a study by Alrahmah et al. [24]. Our results indicate the need to train dental students on following safety protocols in case of needle stick injury.

Overall, students at clinical study level showed a better knowledge regarding HIV transmission routes compared to the preclinical study level (Table 1), which is in agreement with studies by Sallam et al. [25] and Al-Kadhim [26]. Although HIV transmission through saliva in the dental environment has not yet been established [27], only 41% of the dental students in our study were aware of this finding. Similarly, Kumar et al. [13] reported that 56% of dental students were aware of this finding. Our study showed that only 37.5% of the participating dental students answered correctly (no) regarding that cardiopulmonary resuscitation (CPR) can transmit HIV from infected patients. This result is very close to that of Kumar et al. [13] who reported that 28.8% of the surveyed dental students agreed or strongly agreed to do CPR for an HIV patient if needed. Bierrens et al Berden [28] reported that the level of risk for transmission of HIV through CPR is very low and it is estimated to be about one per million resuscitations. The Heart and Stroke Foundation of Canada suggests that the value of CPR outweighs the small, theoretical risk of HIV transmission through CPR [29].

It has been shown that HIV transmission through aerosols is unlikely, requiring the deposition of a sufficient amount of the virus on mucous membranes of the susceptible host [30]. Only 26.4% of participants answered correctly (no) regarding HIV transmission through aerosols by hand pieces, which is lower than reported by Shahzeb et al. (52%) [14] and Alrahmah et al. (70%) [24]. This variation could be due to differences in the sample characteristics and curricula among dental colleges in different countries. Our sample included students in all study levels and intern students, while most published studies excluded preclinical and internship students [13,14,22,24]. One of the surprising findings in our study is that the knowledge of the intern students was lower than the preclinical and clinical students in response to some questions regarding knowledge of HIV. This might be due to the students at the preclinical study level studying the virology course during this level, making their information on HIV fresh during this period.

Unsurprisingly, the knowledge of the internship and clinical students regarding ELISA as a trusted test for HIV infection was far better than the preclinical students (74.5%, 66.2%, and 31.3% respectively), with more females (59.6%) showing knowledge than males (42.8%). The reason for such variation is not clearly understood and warrants further research.

Overall, our results suggest a lower than positive attitude among the participants regarding treatment of HIV patients. This is in line with the results of an earlier study by Yildirim et al. [23] who indicated that the overall attitude of dental students towards HIV patients was 50.91% which is comparable to the studies reported by Fotedar et al. [30] (65.6%) and Sadeghi et al. [31] (57.4%). However, Seacat et al. [32] reported a good overall attitude among dental and dental hygiene students towards HIV patients (81.1%), while Albujeer et al. [33] showed that the level of overall attitude of their dental students was only 21.4%. Our results showed that there were significant differences in attitudes towards HIV patients across study levels of dental students (*p* < 0.05). This was in terms of acceptance to treat HIV-positive patients, ethics of treating HIV-positive patients by dentists, importance of infection control procedures for treatment of HIV patients, routine dental care of HIV patients, dental treatment of all patients including those infected with HIV, and the freedom of the dentist to refuse treatment of HIV-positive patient (Table 3). Notably, overall attitude of the intern and clinical dental students was significantly better compared to the preclinical students.

In 1988, the WHO indicated that all dental professionals must treat HIV patients [34]. Dentists have the ethical obligation to treat HIV patients [35]. Seacat et al. [32] reported that wearing double gloving, refusal to treat a patient, or any biased act toward HIV patients is considered discriminatory. Our results show that only 49.9% of the dental students accepted to treat HIV-positive patients, with intern and clinical dental students showing a more positive attitude than preclinical students, and females showing a better attitude than males (*p* < 0.05). Kumar et al. [13] reported that 45% of the dental students in their study were willing to treat HIV patients, whereas Alsamghan [36] reported 41.3%. Dental educators should take such results into account to promote positive attitudes of dental students towards HIV patients. Kadeh et al. [37] indicated that most dentists in their study refused to treat HIV patients, while Shahzeb et al. [14] reported 24% and Grover et al. [22] reported 21.95%.

Regarding the attitudes of the surveyed dental students toward the importance of infection control procedures during the treatment of HIV patients, the attitude of the preclinical students lags behind the clinical and intern students. While this is not surprising, this result is in line with other previous studies [24,38]. The overall negative attitude could be due to inadequacy or absence of appropriate safety facilities and lack of clear policies in the dental institutions on treatment of HIV patients. We understand that dental students in our study feel uncomfortable to treat HIV patients and fear being infected and might lack knowledge of current infection control guidelines while treating HIV patients.

Our results suggest that dental students in this study have inadequate knowledge regarding oral manifestations of HIV patients (32.7%) (Table 4). Earlier studies identified a better level of knowledge among dental students regarding oral manifestations of HIV patients in comparison with our findings. Singh et al. [11] reported high knowledge among Malaysian dental students of oral manifestation of HIV-positive patients such as oral candidiasis (99.3%), Kaposi’s sarcoma (90.5%), hairy leukoplakia (89.1%), necrotizing ulcerative gingivitis (86.9%), herpes zoster (60.6%), herpes simplex (60.6%), condyloma (29.2%), and salivary gland infection (62.8%). A second study in Malaysia by Al-Kadhim et al. [26] indicated that dental students identified oral candidiasis (97.1%), Kaposi’s sarcoma (94.9%), hairy leukoplakia (91.2%), herpes zoster (56.6%), herpes simplex (64%), salivary gland infection (73.5%), and condyloma (39%). In Turkey, Yildirim et al. [23] reported similar results to our study. They indicated that dental students recognized oral candidiasis (48.43), Kaposi’s sarcoma (33.9%), condyloma (7.97%), xerostomia (22.22%), necrotizing ulcerative gingivitis (23.01%), and salivary gland infection (28.69%) as common oral manifestations in HIV-positive patients. In Indonesia, Wimardhani et al. [20] reported that dental students identified Kaposi’s sarcoma (58%), oral candidiasis (92%), hairy leukoplakia (79.8%), acute necrotizing ulcerative gingivitis (68.4%), herpes simplex (60.9%), and xerostomia (49%).

The low knowledge of dental students in our study regarding oral manifestations of HIV-positive patients might be related to the low number of HIV-positive cases in Saudi Arabia. In 2018, the incidence of HIV in Saudi Arabia was only three cases per 10,000 people. This may explain the relatively negative attitude of dental students in our study regarding the treatment of HIV-positive patients. Remarkably, similar to the studies mentioned earlier, our study indicated that female dental students had better knowledge and attitude towards HIV patients than males. Reasons for such findings are not clear and merit further investigation.

There are a few recommendations that we would like to suggest on the basis of our study. This could include preparing and educating students as early as possible in dental educational institutions at the preclinical study level and at the clinical study level with special programs such as seminars and conferences and others to increase the knowledge of dental students regarding HIV patients. We recommend reinforcing the contribution of dental students in the treatment of HIV-positive patients in the clinical or intern study level under the supervision of specialists and following international infection control protection guidelines to avoid cross-infection with HIV. This would likely enhance the self-confidence of the dental students to overcome the fear and anxiety while treating HIV-positive patients. Future research should be planned to understand in more depth the factors that affected students’ knowledge and attitudes towards HIV patients. This can be a starting point for the promotion of dental students’ perception of HIV patients.

### Limitations

A limitation of this study is the less than optimal response rate that may have biased the findings of this survey. Moreover, there is a difference and variations in the dental curriculum across the Saudi universities. These differences might have biased our results. Additionally, this study is cross-sectional, so we can only analyze the results at one point in time, and thus a longitudinal study could be conducted in the future to observe the change in students’ knowledge and attitude over time. Furthermore, the sample of this study was a random convenience sample of dental students in Saudi Arabia, and this limits the generalizability of the findings.

## 5. Conclusions

The level of knowledge of Saudi dental students about HIV was inadequate and this may be explained by the fact that this disease is uncommon in Saudi Arabia. There were inadequacies in the students’ knowledge regarding safety-related questions about HIV and oral manifestations of HIV. Unfortunately, our dental students showed a negative attitude towards treatment of HIV-positive patients. Intern dental students showed a better knowledge and attitude towards treating HIV patients. The dental school curriculum in Saudi Arabia should, therefore, be updated and improved in order to enhance the students’ knowledge of these aspects of HIV.

We recommend the dental institutions in Saudi Arabia to develop special training programs for dental students regarding HIV. Dental educators and health care planners have an important role in disseminating the knowledge to their students and should consider the findings of this study. Fear and uncertainty towards the treatment of HIV-positive patients and refusal still persist.

## Figures and Tables

**Table 1 healthcare-10-01379-t001:** Characteristics of participants/*n* (%)—(no = 405).

Gender	Age Group	Level of Study
Male, 269 (66.4%) Female, 136 (33.6%)	18–23 years, 285 (71%) 23–30 years, 120 (29%)	Preclinical, 220 (54.3%) Clinical, 130 (32.1%) Intern students, 55 (13.6%)

**Table 2 healthcare-10-01379-t002:** Response of participants to safety-related questions about HIV by gender and study level (no = 405).

Question	% of “Correct” Answers by Study Level and Gender							
	Sample	Study Level			*p*-Value	Gender		*p*-Value
		Preclinical	Clinical	Intern		Male	Female	
1—Do you do HIV test after needle stick injury?	44.2	40.5	48.5	49.1	0.25	46.1	40.4	0.279
2—Do you think that saliva can transmit HIV?	41	41.8	43.1	32.7	0.39	38.7	45.6	0.181
3—Do you think that cardiopulmonary resuscitation can transmit HIV from AIDS patient?	37.5	37	42.3	25.5	0.096	33.8	44.9	0.03 *
4—Can HIV be transmitted through aerosols by hand pieces?	26.4	25.5	25.4	32.7	0.521	29	21.3	0.098
5—Do you think that ELISA is a trusted test for HIV infection?	48.4	31.4	66.2	74.5	<0.001 *	42.8	59.6	0.001 *
**Total**	**39.5**	**35.24**	**45.1**	**42.9**	**0.152**	**38.08**	**42.36**	**0.397**

* Denotes significant difference at *p* < 0.05 as indicated by chi-square statistics.

**Table 3 healthcare-10-01379-t003:** Response of participants to attitude-related questions regarding treatment of HIV patients by gender and study level (no = 405).

Question	% of “Yes” Answers by Study Level and Gender							
	Sample	Study Level			*p*-Value	Gender		*p*-Value
		Preclinical	Clinical	Intern		Male	Female	
1—I accept to treat HIV-positive patients.	49.9	41.8	56.2	67.3	<0.001 *	48	53.7	0.128
2—It is an ethical issue for an HIV-positive patient to be treated by a dentist.	50.9	45	57.7	58.2	<0.001 *	46.8	58.8	0.012 *
3—Infection control procedures are very important for the treatment of HIV patients and take a long time and may affect the work quality of the dentist	57.8	48.2	71.5	63.6	<0.001 *	51.7	69.9	0.001 *
4—Routine dental care should be a part of the treatment of patients with HIV/AIDS	55.1	41.8	70	72.7	<0.001 *	48.7	67.6	0.001 *
5—All dental patients must be treated even if they have HIV/AIDS	36.8	19.5	49.2	76.4	<0.001 *	36.4	37.5	0.061
6—The dentist can make a decision to refuse to treat an HIV-positive patient	50.6	45	59.2	52.7	<0.001 *	46.1	59.6	0.003 *

* Denotes significant difference at *p* < 0.05 as indicated by chi-square statistics.

**Table 4 healthcare-10-01379-t004:** Response of participants to knowledge-related questions regarding oral manifestation of HIV by gender and study level (no = 405).

Question	% of “Correct” Answers by Study Level and Gender							
	Sample	Study Level			*p*-Value	Gender		*p*-Value
		Preclinical	Clinical	Intern		Male	Female	
1—Xerostomia	19.8	24.1	15.4	12.7	0.053	19.7	19.9	0.971
2—Condyloma	20.2	25	11.5	21.8	0.010 *	21.9	16.9	0.235
3—Herpes simplex	16.5	17.3	16.2	14.5	0.879	17.5	14.7	0.476
4—Salivary gland infection	20.7	23.6	20	10.9	0.111	20.1	22.1	0.642
5—Herpes zoster	43.5	26.4	64.6	61.8	<0.001 *	38.7	52.9	<0.001 *
6—Necrotizing ulcerative gingivitis	38.3	14.1	64.6	72.7	<0.001 *	34.6	45.6	0.031 *
7—Hairy leukoplakia	47.4	25.5	65.4	92.7	<0.001 *	44.2	53.7	0.072
8—Oral candidiasis	44.7	20.9	70.8	78.2	<0.001 *	40.9	52.2	0.031 *
9—Kaposi’s sarcoma	43.5	22.3	63.1	81.8	<0.001 *	37.2	55.9	<0.001 *
**Total**	**32.7**	**22.1**	**43.5**	**49.7**	**<0.001 ***	**30.5**	**37.1**	**0.156**

* Denotes significant difference at *p* < 0.05 as indicated by chi-square statistics.

## Data Availability

The questionnaire used in the current study is not publicly available due to certain restrictions. However, it is available from the corresponding author (Bassel Tarakji) on reasonable request.

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
