# Peer review of "Assessment of Knowledge and Attitude of Dental Students towards HIV and Its Oral Manifestations in Saudi Arabia—A Cross-Sectional Study"

_healthcare, 2022, doi:10.3390/healthcare10081379_

Round 1

Reviewer 1 Report

Dear Editor,

Thanks for the opportunity to evaluate the manuscript titled “Assessment of Knowledge and Attitude of Dental Students Towards HIV and Its Oral Manifestations in Saudi Arabia- a Cross-Sectional Study” proposed by Alali and collaborators.

The topic is important and may help improve care to HIV patients in dental clinics. However, the presentation of the data and the overall report was not optimal. Please see my specific comments below:

Abstract

Abstract needs to be polished. There are grammatical mistakes as well as poor word choice or use.

Not sure what “Negative attitude of dental students towards treatment of HIV positive patients.” Is about. So is “Poor knowledge regarding oral manifestations of HIV (32.7%).”

The conclusion does not fit the description of the whole abstract “Intern students showed a better knowledge and attitude towards treating HIV patients compared to preclinical students”: it was not clear that participants were divided into groups.

Introduction

Add references to all claims

There was no need of curriculum and the different levels of dental students in Saudi Arabia. Is it different from those of other countries? This may help the authors present the significance of this study.

The transitions are poor and the logic of the whole section should be revised

Avoid terms such as basically

This section needs to be rewritten, with an improved logic, references as appropriate and English language use.

Materials and Methods

Sample size calculation could be part of the statistical analysis

Were all students included in the study? This should be clarified. If not, what were the inclusion criteria? Were there any exclusion criteria?

The Questionnaire is not well described. Details should be provided.

The overall description does not flow well, because of suboptimal sentence construction and other language issues

Sometimes only p values are provided: the authors should make sure to extract all important data from the tables.

Tables should not have red fonts.

Discussion

This section is lengthy. I would keep it concise. The background information can be reduced.

Clearly discuss your main findings in light of previous studies

Start the section with a brief description of your main findings.

Recommendations should be removed from limitations

Author Response

Reviewer 1:

Dear Editor,

Thanks for the opportunity to evaluate the manuscript titled “Assessment of Knowledge and Attitude of Dental Students Towards HIV and Its Oral Manifestations in Saudi Arabia- a Cross-Sectional Study” proposed by Alali and collaborators.

The topic is important and may help improve care to HIV patients in dental clinics. However, the presentation of the data and the overall report was not optimal. Please see my specific comments below:

Abstract

Abstract needs to be polished. There are grammatical mistakes as well as poor word choice or use.

Not sure what “Negative attitude of dental students towards treatment of HIV positive patients.” Is about. So is “Poor knowledge regarding oral manifestations of HIV (32.7%).”

The conclusion does not fit the description of the whole abstract “Intern students showed a better knowledge and attitude towards treating HIV patients compared to preclinical students”: it was not clear that participants were divided into groups.

Thank you for this comment. The abstract has been revised according to the reviewer’s comments.

Introduction

  • Add references to all claims

Done

  • There was no need of curriculum and the different levels of dental students in Saudi Arabia. Is it different from those of other countries? This may help the authors present the significance of this study.

Thank you very much for this comment. The variations in the dental curriculum across the Saudi Universities and other countries and the impact on dental students’ perception of the HIV patients have been highlighted in the discussion. 

  • The transitions are poor and the logic of the whole section should be revised
  • Avoid terms such as basically
  • This section needs to be rewritten, with an improved logic, references as appropriate and English language use.

The introduction has been revised according to the aforementioned remarks.

 Materials and Methods

  • Sample size calculation could be part of the statistical analysis.

Sample size calculation was part of the planning stage for this research to determine the required sample size. For this reason the authors presented the calculation of the sample size in the methods section.

  • Were all students included in the study? This should be clarified. If not, what were the inclusion criteria? Were there any exclusion criteria?

The targeted sample of this study was undergraduate dental students at government and private Universities in Saudi Arabia including first, second, third, fourth, fifth years, and sixth year in addition to internship dental students at the 2020– 2021 academic year. This has been indicated in the methods section.

  • The Questionnaire is not well described. Details should be provided.

Revised and indicated in red font.

  • The overall description does not flow well, because of suboptimal sentence construction and other language issues.

Revised.

  • Sometimes only p values are provided: the authors should make sure to extract all important data from the tables.

Reviewed and revised.

  • Tables should not have red fonts.

Corrected.

Discussion

This section is lengthy. I would keep it concise. The background information can be reduced.

Clearly discuss your main findings in light of previous studies

Start the section with a brief description of your main findings.

Recommendations should be removed from limitations

The discussion has been revised according to the reviewer’s comments.

Reviewer 2 Report

This is a descriptive study that aims to investigate the knowledge and attitude of Saudi dental students regarding HIV-related aspects. The authors distributed an online questionnaire through social media and WhatsApp.  The results of the study were mainly based on bivariate analysis. The authors found that the level of knowledge and attitude of Saudi dental students regarding HIV-related aspects is inadequate. 

I have some significant concerns regarding the study. First and most importantly, the lack of a valid sampling strategy and information about the colleges and the regions limits the generalisability of the findings and the validity of the study. Second, even if HIV is rare and uncommon in Saudi Arabia, the finding (of inadequate knowledge and attitude) should not be acceptable given the lethality and the implications of the disease. Third, as the study is mainly based on bivariate analysis and the sociodemographic were limited to age, gender and academic year study, the authors did not investigate the factors that could explain the observed associations. 

Author Response

Reviewer 2:

Comments and Suggestions for Authors

This is a descriptive study that aims to investigate the knowledge and attitude of Saudi dental students regarding HIV-related aspects. The authors distributed an online questionnaire through social media and WhatsApp.  The results of the study were mainly based on bivariate analysis. The authors found that the level of knowledge and attitude of Saudi dental students regarding HIV-related aspects is inadequate. 

  • I have some significant concerns regarding the study. First and most importantly, the lack of a valid sampling strategy and information about the colleges and the regions limits the generalisability of the findings and the validity of the study.

Thank you very much for this comment. This limitation has been addressed.

  • Second, even if HIV is rare and uncommon in Saudi Arabia, the finding (of inadequate knowledge and attitude) should not be acceptable given the lethality and the implications of the disease.

The text has been revised to address this comment.

  • Third, as the study is mainly based on bivariate analysis and the sociodemographic were limited to age, gender and academic year study, the authors did not investigate the factors that could explain the observed associations.

We agree with this comment. The study was planned to be a descriptive cross-sectional study. However, we revised the discussion to address this comment: “Future research should be planned to understand in more depth the factors that affected students’ knowledge and attitudes towards HIV patients. This can be a starting point for the promotion of dental students’ perception of the HIV patients”.    

Round 2

Reviewer 2 Report

I would like to thank the authors for incorporating  my comments within the revised version of the study. The authors improved the rationale of the study and addressed its limitations in the discussion. I have no further comments to add.